# Lagrange Optimization of Shock Waves for Two-Dimensional Hypersonic Inlet with Geometric Constraints

Yuling Li [1,2], Lianjie Yue [1,2,*], Chengming He [3], Wannan Wu [1] and Hao Chen [1,*]

1   State Key Laboratory of High Temperature Gas Dynamics, Institute of Mechanics, Chinese Academy of Sciences, Beijing 100190, China
2   University of Chinese Academy of Sciences, No.19(A) Yuquan Road, Beijing 100049, China
3   Wide Range Flight Engineering Science and Application Center, Institute of Mechanics, Chinese Academy of Sciences, Beijing 100190, China
*   Correspondence: yuelj@imech.ac.cn (L.Y.); chenhao@imech.ac.cn (H.C.); Tel.: +86-10-8254-3833 (L.Y.)

**Abstract:** The present paper focuses on the Lagrange optimization of shock waves for a two-dimensional hypersonic inlet by limiting the cowl internal angle and inlet length. The results indicate the significant influences of geometric constraints on the configuration of shock waves and performances of an inlet. Specifically, the cowl internal angle mainly affects the internal compression section; the inlet length affects both the internal and external compression sections where the intensity of internal and external compression shock waves shows a deviation of equal. In addition, the performances of optimized inlets at off-design points are further numerically simulated. A prominent discovery is that a longer inlet favors a higher total pressure recovery at the positive AOA; conversely, a shorter inlet can increase the total pressure recovery at the negative AOA.

**Keywords:** scramjet inlet; Lagrange optimization method; shock configuration





## 1. Introduction

Scramjet was admitted as one of the most promising ways of hypersonic propulsion owing to the advantages of its simple structure and large specific impulse, in which the inlet is a key component and its efficiency is directly of relevance to the performance of scramjets. The total pressure loss related to the thrust loss of the engine [1] and the total pressure recovery is selected as an indicator to evaluate the performance of the inlet in previous studies [2–4]. The shock waves in the scramjet inlet compress the incoming flow and mainly account for the total pressure loss in the inlet [5,6]. Therefore, the optimal configuration of the compression shock waves is a key issue in the inlet design. Generally, the inlet types of scramjets include the axisymmetric inlet [7–11], two-dimensional inlet, sidewall compression inlet [12], and inward-turning inlet [13–15] in which the two-dimensional inlet has been studied substantially [2,3,16–18] because of the advantages of its simple structure, high uniformity of the exit flow field, its ease of integration, and its variable structure design. In addition, the two-dimensional inlet has a representative shock wave configuration, and relevant characteristics could be a good reference for the design of other types of inlets.

Optimization designs on the two-dimensional and three-dimensional inlets have generally been extensively studied by using the response surface method and the genetic algorithm to obtain a higher total pressure recovery and more uniform outflow at the inlet exit [19–21]. The issues affecting the performance of the inlet are considered comprehensively in the above optimization methods, but they are too complicated by consuming a lot of computational resources, making it difficult to reveal the general rules of shock wave configuration. Oswatitsch's optimization design of a shock wave system based on the one-dimensional inviscid assumption was a typical example. He [22] studied the Lagrange

optimization for a two-dimensional supersonic external compression inlet involving multiple oblique shock waves and a normal shock wave at the end and found that the total pressure recovery reaches a peak value when the oblique shocks have the equal intensity. Then, Henderson [23] used the Lagrange optimization method to study the external compression inlet with the end of oblique shock and subsonic outflow. Smart [16] extended the Lagrange optimization method to study the mixed-compression hypersonic inlet where the outflow was supersonic and found that the intensities of internal and external shocks were very close whereas the intensities of oblique shocks were not strictly equal. In addition, Smart found that the total pressure recovery of the inlet can be significantly enlarged by increasing the number of internal compression shock waves from one to two. Referring to Smart's work, Shi et al. [24] and Gu et al. [25] studied the influences of the exit Mach number and focused on the objective of minimum external drag. The results showed that the distribution of optimized shocks intensity is similar to previous studies. Compared with the two- and three-dimensional inlet optimization methods, the one-dimensional optimization method can solve the configuration of the shock wave system within a few seconds and, thereby, accomplish the rapid iterative for inlet design. Most importantly, the one-dimensional inviscid optimization makes it easy to understand the fundamental rules of shock wave configuration.

In practical engineering designs, the situation becomes more complex and generally constrains the cowl internal angle and inlet length, owing to the influences of the external drag and the aircraft layout. However, none of the previous studies based on the Lagrange optimization for a two-dimensional hypersonic inlet have considered the influences of the cowl internal angle and inlet length. Thus, in the present study, we improve the Lagrange optimization by limiting the cowl internal angle and inlet length. The methodology of Lagrange optimization is introduced in Section 2. The influences of shock number, cowl internal angle, and inlet length on the performance of two-dimensional hypersonic inlet are discussed in Section 3, followed by the numerical simulation of the performance of optimized inlets at off-design points in Section 4.

## 2. Methodology of Shock Wave Optimization

### 2.1. Optimization Objective and Geometric Constraints

Figure 1 shows the schematic of a typical two-dimensional scramjet inlet with n compression shock waves, in which $H_1$ and $L$ are the height and length of the inlet, $L_{ex}$ and $L_{in}$ the length of external and internal compression section, and $\theta_c$ the cowl internal angle, respectively. Figure 2 illustrates the schematic of the shock wave system, where the subscript $0, 1, \dots, n - n_{in} - 1$ denotes the number of external shocks, $n_{ex} = n - n_{in}$, the subscript $n - n_{in}, \dots, n - 1$ denotes the number of internal shocks, and $M_0$ to $M_n$ and $p_0$ to $p_n$, respectively, represent the Mach number and static pressure corresponding to each region. It is noted that, in the present study, all concerned flow turning angles $\theta_i$ and corresponding shock angles $\beta_i$ are positive, and, similarly, the variation of the cowl internal angle $\theta_c$ and outflow angle $\theta_e$ in a clockwise direction is considered as positive.

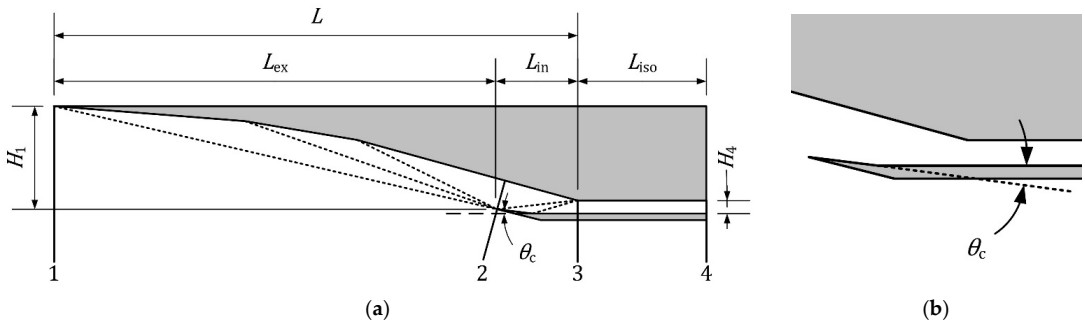

**Figure 1.** Schematic of a two-dimensional scramjet inlet. (**a**) schematic of geometric parameters; (**b**) close-up of the cowl.

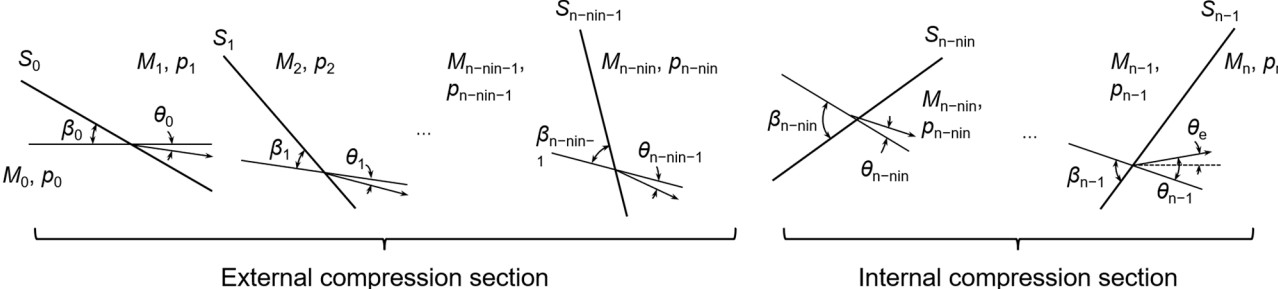

**Figure 2.** Schematic of the shock wave system.

For each oblique shock wave, the total pressure recovery is written as

$$PTR_i = \frac{P_{T_{i+1}}}{P_{T_i}} = f_i^{\gamma/(\gamma-1)} g_i^{\gamma/(\gamma-1)} \tag{1}$$

where

$$f_i = \left(\frac{\gamma-1}{\gamma+1} + \frac{2}{\gamma+1}\frac{1}{M_i^2 \sin^2\beta_i}\right)^{-1} = \frac{\gamma+1}{\gamma-1}\frac{y_i-1}{y_i} \tag{2}$$

$$g_i = \left(\frac{2\gamma}{\gamma+1}M_i^2 \sin^2\beta_i - \frac{\gamma-1}{\gamma+1}\right)^{-1} = \frac{\gamma-1}{\gamma+1}\frac{1}{[4\gamma/(\gamma+1)^2]y_i-1} \tag{3}$$

$$x_i = 1 + [(\gamma-1)/2]M_i^2 \tag{4}$$

$$y_i = 1 + [(\gamma-1)/2]M_i^2 \sin^2\beta_i \tag{5}$$

Therefore, the total pressure recovery of the inlet is calculated as

$$PTR = \prod_{i=0}^{n-1} f_i^{\gamma/(\gamma-1)} g_i^{\gamma/(\gamma-1)} \tag{6}$$

Generally, it contains several constraint conditions. The first condition is Rankine–Hugoniot relation that is given as

$$\psi_i = x_{i+1} - x_i f_i g_i = 0 \quad i = 0,1\ldots\ldots n-1 \tag{7}$$

The second condition is that, after shock waves, the sum of all turning angles is equal to the angle of exit with

$$\psi_n = \sum_{i=n-n_{in}}^{n-1} \theta_i - \sum_{i=0}^{n-n_{in}-1} \theta_i + \theta_e = 0 \tag{8}$$

where $\theta_e$ is the angle of exit, and $\theta_i$ is the turning angle for each single shock that can be expressed by $x_i$ and $y_i$ as

$$\theta_i = \arctan\left[\sqrt{\frac{x_i-y_i}{y_i-1}}\frac{2y_i-(\gamma+1)}{(\gamma+1)x_i-2y_i}\right] \tag{9}$$

The third constraint condition is the parameter of the exit, either for the Mach number with

$$\psi_{n+1} = x_n - x(M_n) = 0 \tag{10}$$

or for the pressure of the exit with

$$\psi_{n+1} = \sum_{i=0}^{n-1} \ln g_i - \ln(PR) = 0 \tag{11}$$

where *PR* denotes the ratio of the exit pressure to the inflow pressure. It is noted that the Mach number constraint is more commonly used in practical scramjet inlet design.

Apart from the above three constraints in previous works, the present paper further considers two more geometric constraints: the inlet length and the cowl internal angle. The inlet length is defined as the distance from the leading edge to the throat shoulder, as shown in Figure 1. To simplify the computation, the concerned multiple internal shocks are assumed to be located at the throat shoulder. Then, as shown in Figure 1, the length of the inlet can be expressed as $L = X_{n-n_{\text{in}}} - X_0$, where $X_0$ and $X_{n-n_{\text{in}}}$ are the coordinates of the leading edge and throat shoulder in x-direction, respectively. We have

$$\psi_{n+2} = X_{n-n_{\text{in}}} - X_0 - L = 0 \tag{12}$$

The relationship between $X_{n-n_{\text{in}}}$ and $x_i, y_i$ should be established to calculate the length of the inlet; however, it cannot be expressed explicitly. Thus, we solve the coordinate of the throat shoulder by using the step marching and normalized the inlet height as 1 to further simplify the calculation. We have

$$X_i = \frac{X_c \tan\left(\sum\limits_{k=0}^{i-1} \theta_k + \delta_i\right) - Y_c + Y_{i-1} - X_{i-1} \tan\left(\sum\limits_{k=0}^{i-1} \theta_k\right)}{\tan\left(\sum\limits_{k=0}^{i-1} \theta_k + \delta_i\right) - \tan\left(\sum\limits_{k=0}^{i-1} \theta_k\right)} \quad i = 1, 2, \ldots\ldots, n - n_{\text{in}} \tag{13}$$

$$Y_i = Y_c + (X_i - X_c) tg\left(\sum\limits_{k=0}^{i-1} \theta_k + \delta_i\right) \quad i = 1, 2, \ldots\ldots, n - n_{\text{in}} \tag{14}$$

where

$$\delta_i = \begin{cases} \beta_i \ i = 1, 2, \ldots\ldots, n - n_{\text{in}} - 1 \\ \pi - \beta_i \ i = n - n_{\text{in}} \end{cases} \tag{15}$$

$$\beta_i = \arcsin\left(\sqrt{(y_i - 1)/(x_i - 1)}\right) \tag{16}$$

By assuming $X_c = Y_c = 0$, Equations (13) and (14) can be simplified as

$$X_i = \frac{Y_{i-1} - X_{i-1} \tan\left(\sum\limits_{k=0}^{i-1} \theta_k\right)}{\tan\left(\sum\limits_{k=0}^{i-1} \theta_k + \delta_i\right) - \tan\left(\sum\limits_{k=0}^{i-1} \theta_k\right)} \quad i = 1, 2, \ldots\ldots, n - n_{\text{in}} \tag{17}$$

$$Y_i = X_i \tan\left(\sum\limits_{k=0}^{i-1} \theta_k + \delta_i\right) \tag{18}$$

$$X_0 = -\arctan(\beta_0) \ y_0 = -1 \tag{19}$$

Furthermore, the constraint of cowl internal angle has the geometry relationship

$$\psi_{n+3} = \sum\limits_{i=n-n_{\text{in}}-1}^{n-1} \theta_i - \theta_c + \theta_e = 0 \tag{20}$$

### 2.2. Lagrange Optimization Algorithm

The Lagrange algorithm is used to solve the optimization problem with multiple constraints. To obtain the extremum of the objective function $G(x_1, x_2, \ldots, x_n, y_0, y_1, \ldots, y_{n-1})$ with m constraints, the optimization function can be established as

$$F = G + \sum\limits_{i=0}^{m} \lambda_i \psi_i \tag{21}$$

This optimization function has 2n + m + 1 independent variables of $x_1, x_2, \ldots \ldots, x_n$, $y_0, y_1, \ldots \ldots, y_{n-1}$, and $\lambda_0, \lambda_1, \ldots \ldots, \lambda_m$, and setting its derivative with respect to the independent variable leads to

$$
\begin{aligned}
\frac{\partial F}{\partial x_i} &= 0 \quad i = 1, 2, \ldots \ldots n \\
\frac{\partial F}{\partial y_i} &= 0 \quad i = 0, 1, \ldots \ldots n - 1 \\
\frac{\partial F}{\partial \lambda_i} &= 0 \quad i = 0, 1, \ldots \ldots m
\end{aligned}
\tag{22}
$$

The extremum of *G* can be obtained by solving the equation set immediately.

For the specific optimization problem in the present paper, the optimization function is

$$
\begin{aligned}
F &= \sum_{i=0}^{n-1} (\gamma \ln f_i + \ln g_i) + \sum_{i=0}^{n-1} \lambda_i (x_{i+1} - x_i f_i g_i) + \lambda_n \left( \sum_{i=n-n_{in}}^{n-1} \theta_i - \sum_{i=0}^{n-n_{in}-1} \theta_i + \theta_e \right) \\
&\quad + \lambda_{n+1}(x_n - x_e) + \lambda_{n+2}(X_{n-n_{in}} - X_0 - L) + \lambda_{n+3} \left( \sum_{i=n-n_{in}-1}^{n-1} \theta_i - \theta_c + \theta_e \right)
\end{aligned}
\tag{23}
$$

where the logarithm of the total pressure recovery is selected as the objective function as

$$
G = (\gamma - 1)\ell n(PT) = \sum_{i=0}^{n-1} (\gamma \ell n f_i + n g_i)
\tag{24}
$$

In the logarithm form, $f_i$ and $g_i$ are decoupled to facilitate subsequent derivations. The derivative of the Equation (23) with respect to $x_1, x_2, \ldots \ldots, x_n$ is

$$
\begin{aligned}
\lambda_{i-1} - \lambda_i f_i g_i - \lambda_n \frac{\partial \theta_i}{\partial x_i} + \lambda_{n+2} \frac{\partial X_{n-n_{in}}}{\partial x_i} &= 0 \quad i = 0, 1, \ldots \ldots n - n_{in} - 1 \\
\lambda_{i-1} - \lambda_i f_i g_i + \lambda_n \frac{\partial \theta_i}{\partial x_i} + \lambda_{n+2} \frac{\partial X_{n-n_{in}}}{\partial x_i} &= 0 \quad i = n - n_{in} \\
\lambda_{i-1} - \lambda_i f_i g_i + \lambda_n \frac{\partial \theta_i}{\partial x_i} + \lambda_{n+2} \frac{\partial X_{n-n_{in}}}{\partial x_i} + \lambda_{n+3} \frac{\partial \theta_i}{\partial x_i} &= 0 \quad i = n - n_{in} + 1, \ldots \ldots n - 1 \\
\lambda_{n-1} + \lambda_{n+1} + \lambda_{n+2} \frac{\partial X_{n-n_{in}}}{\partial x_i} &= 0 \quad i = n
\end{aligned}
\tag{25}
$$

and its derivative with respect to $y_0, y_1, \ldots \ldots, y_{n-1}$ is

$$
\begin{aligned}
\frac{\gamma}{f_i}\frac{df_i}{dy_i} + \frac{1}{g_i}\frac{dg_i}{dy_i} - \lambda_i x_i \left( f_i \frac{dg_i}{dy_i} + g_i \frac{df_i}{dy_i} \right) - \lambda_n \frac{\partial \theta_i}{\partial y_i} + \lambda_{n+2}\frac{\partial X_{n-n_{in}}}{\partial y_i} &= 0 \quad i = 0, 1, \ldots \ldots n - n_{in} - 1 \\
\frac{\gamma}{f_i}\frac{df_i}{dy_i} + \frac{1}{g_i}\frac{dg_i}{dy_i} - \lambda_i x_i \left( f_i \frac{dg_i}{dy_i} + g_i \frac{df_i}{dy_i} \right) - \lambda_n \frac{\partial \theta_i}{\partial y_i} + \lambda_{n+2}\frac{\partial X_{n-n_{in}}}{\partial y_i} &= 0 \quad i = n - n_{in} \\
\frac{\gamma}{f_i}\frac{df_i}{dy_i} + \frac{1}{g_i}\frac{dg_i}{dy_i} - \lambda_i x_i \left( f_i \frac{dg_i}{dy_i} + g_i \frac{df_i}{dy_i} \right) + \lambda_n \frac{\partial \theta_i}{\partial y_i} + \lambda_{n+2}\frac{\partial X_{n-n_{in}}}{\partial y_i} + \lambda_{n+3}\frac{\partial \theta_i}{\partial y_i} &= 0 \quad i = n - n_{in} + 1, \ldots \ldots n - 1
\end{aligned}
\tag{26}
$$

where

$$
\frac{\partial \theta_i}{\partial x_i} = \frac{1}{1 + \frac{(x_i - y_i)(2y_i - \gamma - 1)^2}{(y_i - 1)((\gamma+1)x_i - 2y_i)^2}} \left\{ \frac{1}{2} \frac{(2y_i - \gamma - 1)}{\sqrt{\frac{x_i - y_i}{y_i - 1}}((\gamma+1)x_i - 2y_i)(y_i - 1)} - \frac{\sqrt{\frac{x_i - y_i}{y_i - 1}}(2y_i - \gamma - 1)(\gamma + 1)}{((\gamma+1)x_i - 2y_i)^2} \right\}
\tag{27}
$$

$$
\frac{\partial \theta_i}{\partial y_i} = \frac{1}{1 + \frac{(x_i - y_i)(2y_i - \gamma - 1)^2}{(y_i - 1)((\gamma+1)x_i - 2y_i)^2}} \left\{ \frac{1}{2} \frac{(2y_i - \gamma - 1)\left[ -\frac{1}{y_i - 1} - \frac{x_i - y_i}{(y_i - 1)^2} \right]}{\sqrt{\frac{x_i - y_i}{y_i - 1}}((\gamma+1)x_i - 2y_i)} + \frac{2\sqrt{\frac{x_i - y_i}{y_i - 1}}}{(\gamma+1)x_i - 2y_i} + \frac{2\sqrt{\frac{x_i - y_i}{y_i - 1}}(2y_i - \gamma - 1)}{((\gamma+1)x_i - 2y_i)^2} \right\}
\tag{28}
$$

In Equation (26), $\frac{\partial X_{n-n_{in}}}{\partial x_i}$ can be solved by step marching as

$$\frac{\partial X_i}{\partial x_j} = \frac{\frac{\partial Y_{i-1}}{\partial x_i} - \frac{\partial X_{i-1}}{\partial x_i}\tan\left(\sum\limits_{k=0}^{i-1}\theta_k\right) - X_{i-1}\sec^2\left(\sum\limits_{k=0}^{i-1}\theta_k\right)\sum\limits_{k=0}^{i-1}\left(\frac{\partial\theta_k}{\partial x_i}\right)}{tg\left(\sum\limits_{k=0}^{i-1}\theta_k+\delta_i\right) - \tan\left(\sum\limits_{k=0}^{i-1}\theta_k\right)}$$
$$- \frac{Y_{i-1} - X_{i-1}\tan\left(\sum\limits_{k=0}^{i-1}\theta_k\right)}{\left[\tan\left(\sum\limits_{k=0}^{i-1}\theta_k+\delta_i\right) - \tan\left(\sum\limits_{k=0}^{i-1}\theta_k\right)\right]^2}\left[\sec^2\left(\sum\limits_{k=0}^{i-1}\theta_k+\delta_i\right)\left(\sum\limits_{k=0}^{i-1}\left(\frac{\partial\theta_k}{\partial x_i}\right)+\frac{\partial\delta_i}{\partial x_i}\right) - \sec^2\left(\sum\limits_{k=0}^{i-1}\theta_k\right)\sum\limits_{k=0}^{i-1}\left(\frac{\partial\theta_k}{\partial x_i}\right)\right]$$

(29)

$$\frac{\partial Y_i}{\partial x_j} = \frac{\partial X_i}{\partial x_i}\tan\left(\sum_{k=0}^{i-1}\theta_k+\delta_i\right) + X_i\sec^2\left(\sum_{k=0}^{i-1}\theta_k+\delta_i\right)\left(\sum_{k=0}^{i-1}\left(\frac{\partial\theta_k}{\partial x_i}\right)+\frac{\partial\delta_i}{\partial x_i}\right)$$

(30)

$$\frac{\partial X_i}{\partial y_j} = \frac{\frac{\partial Y_{i-1}}{\partial y_i} - \frac{\partial X_{i-1}}{\partial y_i}tg\left(\sum\limits_{k=0}^{i-1}\theta_k\right) - X_{i-1}\sec^2\left(\sum\limits_{k=0}^{i-1}\theta_k\right)\sum\limits_{k=0}^{i-1}\left(\frac{\partial\theta_k}{\partial x_i}\right)}{tg\left(\sum\limits_{k=0}^{i-1}\theta_k+\delta_i\right) - tg\left(\sum\limits_{k=0}^{i-1}\theta_k\right)}$$
$$- \frac{Y_{i-1} - X_{i-1}\tan\left(\sum\limits_{k=0}^{i-1}\theta_k\right)}{\left[\tan\left(\sum\limits_{k=0}^{i-1}\theta_k+\delta_i\right) - \tan\left(\sum\limits_{k=0}^{i-1}\theta_k\right)\right]^2}\left[\sec^2\left(\sum\limits_{k=0}^{i-1}\theta_k+\delta_i\right)\left(\sum\limits_{k=0}^{i-1}\left(\frac{\partial\theta_k}{\partial y_i}\right)+\frac{\partial\delta_i}{\partial y_i}\right) - \sec^2\left(\sum\limits_{k=0}^{i-1}\theta_k\right)\sum\limits_{k=0}^{i-1}\left(\frac{\partial\theta_k}{\partial y_i}\right)\right]$$

(31)

$$\frac{\partial Y_i}{\partial y_j} = \frac{\partial X_i}{\partial y_i}\tan\left(\sum_{k=0}^{i-1}\theta_k+\delta_i\right) + X_i\sec^2\left(\sum_{k=0}^{i-1}\theta_k+\delta_i\right)\left(\sum_{k=0}^{i-1}\left(\frac{\partial\theta_k}{\partial y_i}\right)+\frac{\partial\delta_i}{\partial y_i}\right)$$

(32)

when $k \neq i$, $\frac{\partial\theta_k}{\partial y_i} = 0$

$$\frac{\partial X_0}{\partial x_j} = \frac{2}{tg^2(\beta_0)}\frac{\partial\beta_0}{\partial x_j} \quad \frac{\partial X_0}{\partial y_j} = \frac{2}{tg^2(\beta_0)}\frac{\partial\beta_0}{\partial y_j} \quad \frac{\partial Y_0}{\partial x_j} = \frac{\partial Y_0}{\partial y_j} = 0$$

(33)

$$\frac{\partial\beta_i}{\partial x_i} = -\frac{1}{2}\frac{\sqrt{(y_i-1)/(x_i-1)}}{(x_i-1)\sqrt{1-(y_i-1)/(x_i-1)}}$$

(34)

$$\frac{\partial\beta_i}{\partial y_i} = \frac{1}{2}\frac{1}{\sqrt{(y_i-1)/(x_i-1)}(x_i-1)\sqrt{1-(y_i-1)/(x_i-1)}}$$

(35)

Consequently, a system of $3n + 4$ equations is established with the same number of independent variables. We can obtain the optimized geometry configuration of the two-dimensional scramjet inlet by solving the equation system.

## 3. Results and Discussion

### 3.1. Characteristics of Inlet without Limiting Cowl Internal Angle and Length

This section discusses a benchmark case of the inlet design without limiting the cowl internal angle and the length under the design condition of $\gamma = 1.4$, $M_0 = 6$, $M_e = 2.7$, and $\theta_e = 0°$. The number of external shocks varies from one to six, and the number of internal shocks varies from one to three.

Figure 3 shows the influences of the number of shocks on the total pressure recovery, cowl internal angle and length, respectively. The total pressure recovery increases as the number of internal or external shocks increases. The increment of total pressure recovery at a small shock number is larger than that at a large shock number. Similarly, the cowl internal angle increases slightly with the increase of the external shock number but decreases substantially as the internal shock number increases. The inlet length is lengthened by increasing the external shock number because of the reduction of the angle of the first shock, whereas is nearly unchanged with varying the internal shock number.

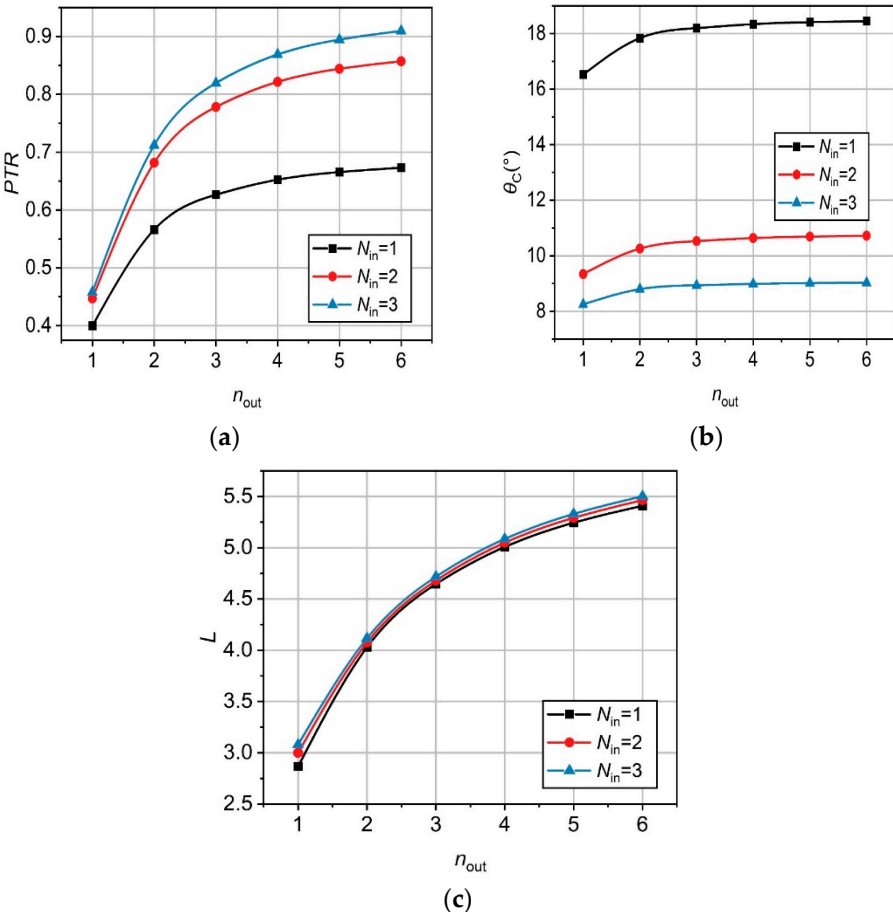

**Figure 3.** Influences of the number of shock waves on (**a**) the total pressure recovery; (**b**) the cowl internal angle; and (**c**) the inlet length.

Based on the results of total pressure recovery, cowl internal angle, and inlet length in Figure 3, a representative case with three external shocks and two internal shocks is considered as the benchmark case for the comparison study as follows. The detailed parameters are listed in Table 1. In addition, it has $\theta_c = \theta_4 = 10.52°$ when $\theta_e = 0$ and $n_{in} = 2$

**Table 1.** Geometric parameters of the benchmark inlet.

| Parameter | Value | Parameter | Value |
|:---:|:---:|:---:|:---:|
| $L$ | 4.68 | $\theta_0/°$ | 5.63 |
| $L_{ex}$ | 4.11 | $\theta_1/°$ | 6.45 |
| $L_{in}$ | 0.57 | $\theta_2/°$ | 7.41 |
| $PTR/\%$ | 77.8 | $\theta_3/°$ | 8.96 |
| $ICR$ | 2.75 | $\theta_4/°$ | 10.52 |

### 3.2. The Influence of the Limit of Cowl Internal Angle

In a practical inlet, the external drag increases with the increase of the cowl internal angle. Thus, we must choose a reasonable cowl internal angle to reduce the external drag. We use the same parameters in the benchmark case by involving three external shocks and two internal shocks and by varying the cowl internal angles among 1, 3, 5, 7, 9, 12, and 14° with free inlet length.

Figure 4a shows the evolution of total pressure recovery and internal compression ratio (ICR) with varying cowl internal angle $\theta_c$, in which ICR is defined as the area ratio of profile 2 to profile 3, as shown in Figure 1. Generally, a larger ICR tends to lead to enhanced

compression ability but more easily allows inlet unstart [5,26], which results in serious damages for the whole engine [27–31]. It is clearly seen that, the variation curves of total pressure recovery and ICR are nearly overlapped in Figure 4a, showing a non-monotonic variation with $\theta_c$ and reaching a peak value of about 77.8% for total pressure recovery and 2.75 for ICR at $\theta_c = 10.53°$. Specifically, the pressure recovery is 88.5%, 93.2%, and 97.0%, respectively, to the maximum total pressure recovery at the cowl internal angles of 3, 5, and 7°.

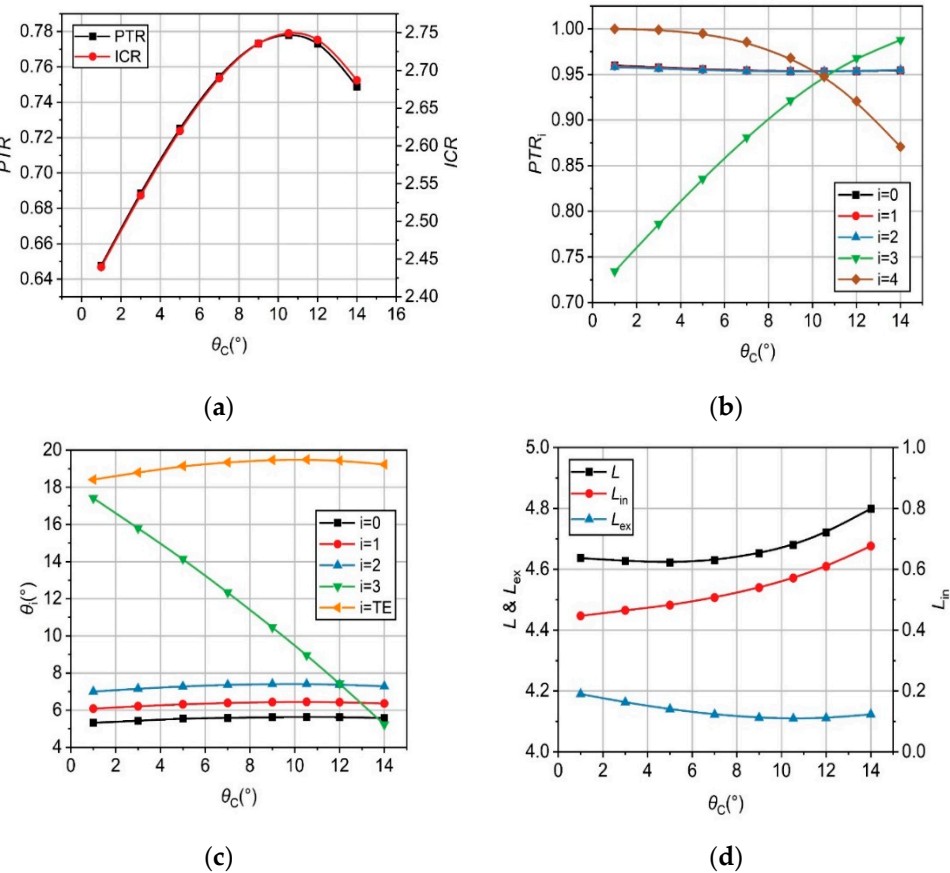

**Figure 4.** Influences of the cowl internal angle on (**a**) the total pressure recovery and IC; (**b**) the pressure recovery of a single shock; (**c**) the flow turning angle; and (**d**) the inlet length and length distribution.

Figure 4b shows the evolution of the pressure recovery of a single shock with the cowl internal angles. Smart's study on hypersonic mixed-compression inlets indicated that the intensities of internal or external shocks were, respectively, approximately equal in the optimized inlet. As shown in Figure 4b, the pressure recovery of three external shocks, $PTR_0$, $PTR_1$, and $PTR_2$, are nearly same and slightly changed with varying the cowl internal angle, regardless of the limitation of the cowl internal angle. In addition, Figure 4c shows the flow turning angle of a single shock with varying the cowl internal angle. As the cowl internal angle deviated from the benchmark inlet, the flow turning angle tends to decrease slightly. The influences of the cowl internal angle limitation are reflected mainly in the internal compression section. When the flow direction of the exit remains constant, the intensity of the first and the second internal shocks is not equal. As shown in Figure 4b, the difference between $PTR_3$ and $PTR_4$ becomes increasingly enlarged when the cowl internal angle deviates from the reference value of 10.53°, leading to decreased total pressure recovery. Meanwhile, the flow turning angle of external shocks tends to decrease so as to compensate for the total pressure loss in the internal section, but the compensation is tiny.

In addition, similar to the influence of the cowl internal angle on the total pressure recovery that mainly reflected in the internal compression section, the ICR depends on the structure of the internal compression section. This is why the same variation trend of ICR and total pressure recovery occurs in Figure 4a. The deviation of the cowl internal angle to the reference value can reduce the total pressure recovery, but the ICR decreases correspondingly, which helps the start of the inlet.

Figure 4d shows the influences of cowl internal angle on inlet length and length distribution. The length of the external compression section $L_{ex}$ is lengthened by the deviation of the cowl internal angle to the reference value slightly, and the difference between the maximum and minimum was only 1.9%, which corresponds to the variation trend of the flow turning angle in Figure 4c. Different from the external compression section, the length of the internal section $L_I$ always increases with the increase of the cowl internal angle. It is seen that the length of the internal section is equal to the horizontal projected length of the fourth shock in Figure 1. When the total flow turning angle of external shocks has slight variation, the fourth shock decreases as the cowl internal angle decreases, and the corresponding shock angle decreases, leading to a decrease of the included angle of the shock and horizontal direction $A_3 = |\beta_3 - \theta_{TE}|$, as shown in Figure 5. Therefore, the horizontal projected length of the fourth shock increases. As shown in Figure 4d, when the cowl internal angle is smaller than the reference value, it has little influence on inlet length. In a usual case, the cowl internal angle is limited to a small angle to reduce the external drag; therefore, the influence of the cowl internal angle on length can be ignored.

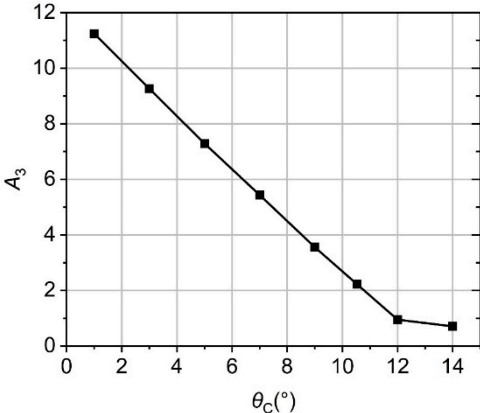

**Figure 5.** Absolute value of the angle between the fourth shock and the horizontal direction.

### 3.3. Influence of the Limit of Inlet Length

In this section, we use the same parameters as the benchmark case and study the influence of inlet length on the performance of inlet with fixed cowl internal angle. Figure 6a shows the variation of total pressure recovery and ICR with inlet length. It is seen that as the inlet length deviated from the reference value of 4.68, both the total pressure recovery and the ICR of the inlet decreases. Figure 6b shows the length change of the internal and external compression sections with the inlet limit length. A prominent linear change of the length in the external compression section is observed; conversely, the change of the length in the internal compression section is negligible. Furthermore, by comparing Figures 4a and 6a, the influence of the inlet length on the internal contraction ratio is much weaker than that of the internal cowl angle.

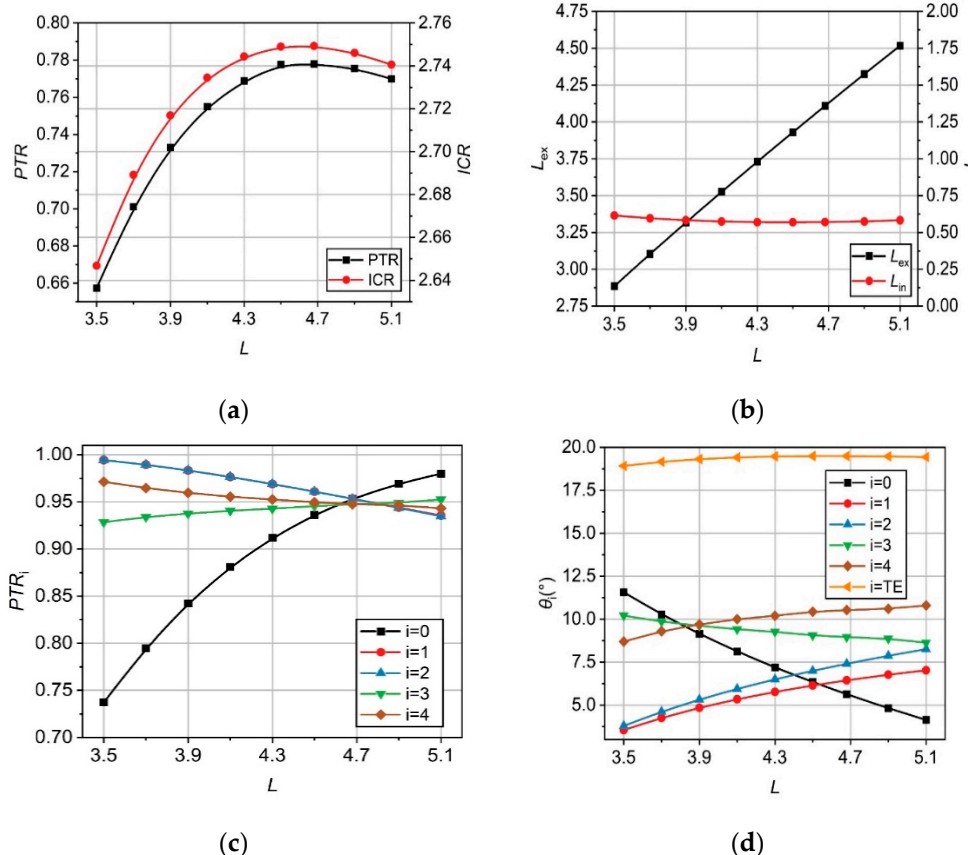

**Figure 6.** Influences of the length limit on (**a**) the total pressure recovery and ICR; (**b**) the length distribution; (**c**) the total pressure recovery of a single shock; and (**d**) the flow turning angle.

Figure 6a,b show the total pressure recovery and flow turning angle varying with the inlet length, respectively. As shown in Figure 6d, based on the relation of $L_{ex} = H_1 / \tan \beta_0$, the increase of inlet length tends to reduce $\beta_0$ and $\theta_0$, and, thereby, decrease the intensity of the first external shock. To compensate for the reduced intensity of the first shock, the intensities of the second and third external shocks increase and the corresponding total pressure recovery coefficients ($PTR_1$ and $PTR_2$) decrease, in which the intensity of the two shocks appears approximately the same as increasing the inlet length. The difference between the first external shock and the second and third one can be enlarged by increasingly deviating the inlet length away from the reference value of 4.68, which explains why the length limit leads to a drop in total pressure recovery. In addition, Figure 6d also shows that length limit significantly influences the external shocks; however, it cannot cause a big change on the total external flow turning angle. The cowl internal angle $\theta_c$ and the flow angle of the last shock $\theta_4$ have the same value increasing with the increase of inlet length.

Furthermore, as shown in Figure 6c, increasing the inlet length shows an increase of the intensity of the first internal shock and a decrease of the counterpart for the second internal shock. Generally, the shocks with approaching intensities tend to lead to a larger total pressure recovery. However, this rule is seemingly not satisfied in the present optimized results. This can be understood in the two examples with the inlet length of 3.5 and 5.1 shown in Table 2. It compares the configurations between the ideal situation of equal intensity and the optimized configuration. It is seen that when the inlet length is smaller than the reference value, the equal intensity configuration requires a longer length of the internal compression section $L_{in}$, and, thereby, reduces the length of the external compression section $L_{ex}$ and enhances the intensity of the first external compression shock wave, further leading to a decrease in the intensity of the second and third external compression

shock waves. As a result, the intensity difference of three external compression shocks is amplified, and the pressure recovery of external compression section PTRex decreases. The decrement is larger than the increment caused by the equal intensity of internal shocks, leading to the decrease of the total pressure recovery. Similarly, when the inlet length is larger than the reference value, the equal intensity configuration would increase the length of the external compression section and intensity difference of the external compression shock wave but decrease the total pressure recovery of the external pressure section.

**Table 2.** Comparison between equal configuration intensity and optimized configuration.

| $L$ | | $PTR$ | $PTR_0$ | $PTR_1$ | $PTR_2$ | $PTR_{ex}$ | $PTR_3$ | $PTR_4$ | $PTR_{in}$ | $L_{in}$ | $L_{ex}$ |
|---|---|---|---|---|---|---|---|---|---|---|---|
| 3.5 | Opt. | 0.6574 | 0.7373 | 0.9943 | 0.9944 | 0.7289 | 0.9285 | 0.9713 | 0.9018 | 0.6148 | 2.8852 |
| | Equal | 0.6520 | 0.7257 | 0.9951 | 0.9951 | 0.7186 | 0.9530 | 0.9530 | 0.9082 | 0.6572 | 2.8428 |
| 5.1 | Opt. | 0.7699 | 0.9799 | 0.9354 | 0.9348 | 0.8569 | 0.9526 | 0.9432 | 0.8985 | 0.5826 | 4.5174 |
| | Equal | 0.7698 | 0.9802 | 0.9351 | 0.9344 | 0.8565 | 0.9481 | 0.9480 | 0.8988 | 0.5757 | 4.5243 |

### 3.4. Numerical Study of Performances at Off-Design Conditions

Tests of an inlet at the off-design conditions are of significance to evaluate its performances for the practical flight. In this section, two important characteristics of the total pressure recovery and the flow capture performance for the optimized inlets in Sections 3.2 and 3.3 are numerically studied under different off-design conditions.

#### 3.4.1. Performances at Difference Mach Number

Figure 7 shows the influences of cowl internal angle and length limit on mass capture ratio at off-designed Mach number with the angle of attack (AOA) of zero. The mass capture ratio is defined as the mass flux that actually enters the cowl that normalized by the theoretically calculated mass flow rate across the inlet based on the capture area [32]. As shown in Figure 7a, the cowl internal angle has little influences on the mass capture ratio under the off-designed Mach number. This is because the cowl internal angle mainly affects the internal compression section but not the flow and shocks in the external compression section that accounting for the variation of the mass capture ratio. As shown in Figure 7b, the mass capture ratio shows a prominent decrease with the inlet length at off-design Mach number, and the decrement of mass capture ratio mainly occurs at small inlet length. In other words, the influence of length variation on mass capture ratio is very small near the reference length. Therefore, from the perspective of mass capture ratio, unless the inlet is very short, the influence of the cowl internal angle and inlet length limit on the mass capture ratio at off-designed Mach number is not necessarily considered for the inlet design.

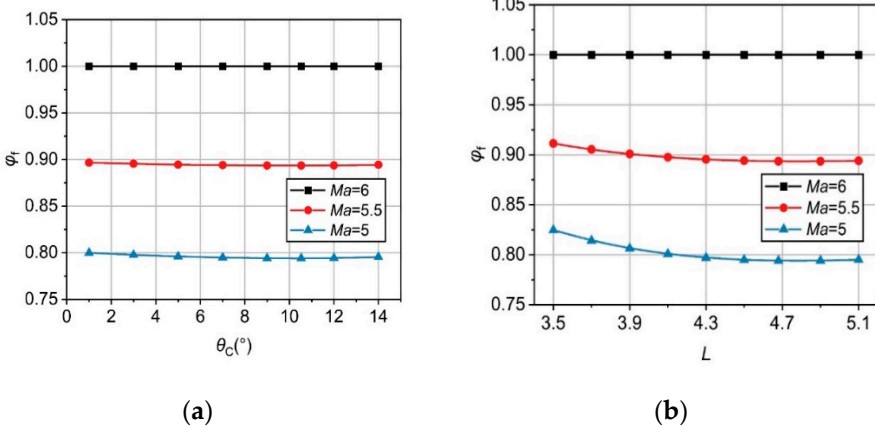

(**a**)  (**b**)

**Figure 7.** Evolution of the mass capture ratio at off-design Mach numbers with varying (**a**) the cowl internal angle; and (**b**) the inlet length.

### 3.4.2. Performances at Different AOA

Figure 8a shows the influence of the cowl internal angle on the pressure recovery under the off-design AOA. It is seen that the deviation of AOA from the design value of zero always reduces the total pressure recovery. As the AOA varyies from −9 to 9°, the peak values of the total pressure recovery are respectively $\theta_c$ = 7, 9, 9, 10.53, 10.53, 10.53 and 12°, showing that the critical cowl internal angle occurring maximum total pressure recovery increases as the AOA increases. When comparing two curves of the positive and negative AOA with the same absolute value, the variation trend of total pressure recovery with the cowl internal angle is not exactly the same, but there is no obvious difference.

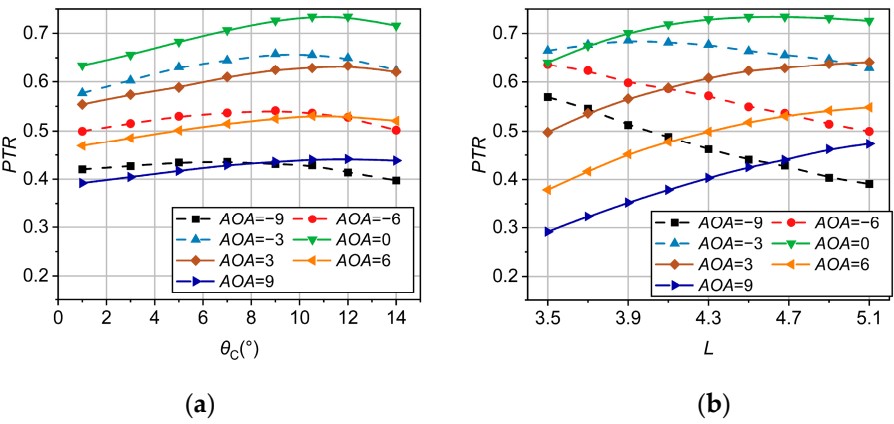

(**a**)                                         (**b**)

**Figure 8.** Evolution of the total pressure recovery at off-design AOA with varying (**a**) the inlet length; and (**b**) the cowl internal angle.

Figure 8b shows the influence of inlet length on pressure recovery under off-design AOA. When the AOA is positive, the increase of the AOA always results in a decrease of the pressure recovery, which is caused by the increasing loss of the first shock. For the negative AOA, the total pressure recovery decreases with decreasing AOA; however, when the length is less than or equal to 3.7, the total pressure recovery increases first and then decreases as the AOA decreases. For the negative AOA, the first shock is weakened, and that increases the total pressure recovery; however, the Mach number after the shock increases and enhances the total pressure loss of downstream shocks. The total pressure recovery of the entire inlet relies on the competition between the above two aspects. For a very short inlet, the first flow turning angle is large, leading to an enhanced augmentation of the first shock wave at the negative AOA. In addition, for the positive and the negative AOA with the same absolute value, it shows the opposite variation of the total pressure recovery with length. At the negative AOA, a shorter inlet length can lead to a higher total pressure recovery, whereas at the positive AOA, the total pressure recovery always increases as increasing length, which is also caused by the shock wave approaching or deviating from the equal intensity distribution.

Figure 9 shows the influence of the cowl internal angle and the length limit on the mass capture ratio at AOA. It is seen that the mass capture ratio always increases with the increase of the AOA, owing to the increase of the inlet windward area. For a specific AOA, the mass capture ratio is nearly uninfluenced by varying the cowl internal angle, which is because the cowl internal angle mainly affects the internal compression section but not the flow and shocks in the external compression section as discussed before. Different from the cowl internal angle, the mass capture ratio shows a linear increase with the inlet length at the positive AOA and a linear decrease with the inlet length at the negative AOA. Furthermore, in a specific range of AOA, a long inlet has a longer external compression section, resulting in a larger change in the windward area and a more prominent change in mass capture ratio.

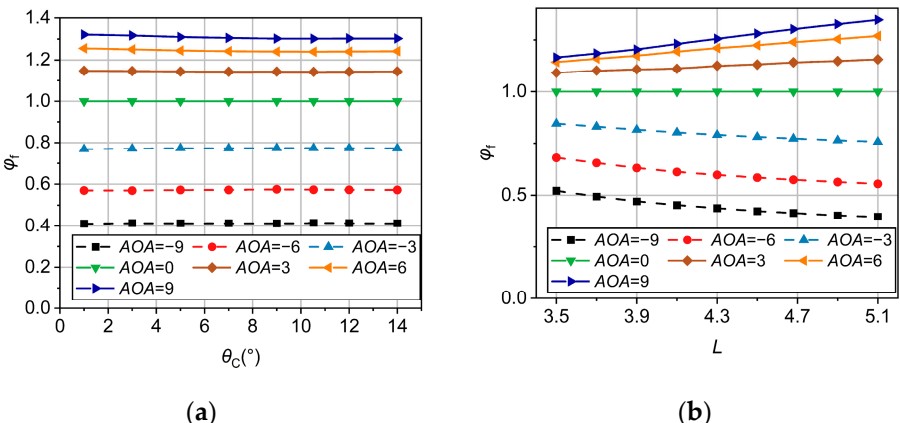

(a)                  (b)

**Figure 9.** Evolution of the mass capture ratio at off-design AOA with varying (**a**) the cowl internal angle; and (**b**) the cowl internal angle.

From Figures 7–9, it is seen that under various different off-design conditions, the cowl internal angle has little influence on the mass capture ratio and affects the total pressure recovery in a similar manner when compared to that of the design conditions. The inlet length has little influence on the mass capture ratio at off-design Mach number but has significant influences on the inlet performance at other off-design conditions. For either the total pressure recovery or mass capture ratio, a shorter inlet length can improve the performance at the negative AOA, while a longer inlet can improve the performance at the positive AOA.

## 4. Conclusions

The present work proposes an inviscid Lagrange optimization method for the two-dimensional inlet by considering two geometric constraints of the cowl internal angle and inlet length for practical inlet design, and further studies the performances of inlet that are influenced by the cowl internal angle and inlet length at the design point and off-design point, respectively.

1. A benchmark case of the inlet without geometric constraints based on the Lagrange optimization method indicates that the total pressure recovery varies non-monotonically with the cowl internal angle and reaches the peak value at the reference cowl internal angle. The influences of the cowl internal angle that deviated from the reference value are mainly reflected in the internal compression section by slightly decreasing the flow turning angle of external shocks. The shock configuration rule of equal intensity distribution is still satisfied by varying the cowl internal angle, which accounts for the negligible influences of the cowl internal angle on the inlet length.
2. The influences of the inlet length that deviate from the reference value can be observed in both internal and external sections and lead to a decrease of the total pressure recovery. For the internal compression section, the influences of the inlet length are larger than that of the cowl internal angle. The shock configuration rule of equal intensity distribution is not satisfied by varying the inlet length.
3. For the optimized inlet by limiting the cowl internal angle, it appears to be nearly the same evolutions of the total pressure recovery with the cowl internal angle at design or off-design Mach number. The total pressure recovery decreases as the cowl internal angle deviates from the reference value that can be observed for both the designed or off-designed AOA, in which the critical value of the cowl internal angle increases with AOA. The mass capture ratio is slightly influenced owing to the fact that external shocks are not sensitive to the change of the cowl internal angle.
4. For the optimized inlet by limiting the inlet length, the total pressure recovery decreases with the deviation of inlet length from the reference value. At the non-designed AOA, it shows an opposite variation of the total pressure recovery with inlet length.

Specifically, at the positive AOA, a longer inlet can improve the performance; whereas at the negative AOA, a shorter inlet is expected to increase the total pressure recovery. In addition, the increase of inlet length would cause a prominent decrease of the mass capture ratio at the Mach number lower than the design value only for a shorter inlet. The mass capture ratio changes noticeably for a long inlet because the windward area can be changed easily.

It is worth mentioning that the present paper was carried out in a two-dimensional inviscid framework. In the actual 2D inlet, the 3D effects brought by the side wall and the shock boundary layer interactions tend to decrease the total pressure recovery. However, our inviscid theory is still valid because shock waves are the primary source of total pressure loss. Certainly, the 3D effects and boundary layer merits further studies to improve the accuracy of the theory.

**Author Contributions:** Conceptualization, Y.L. and L.Y.; methodology, Y.L. and L.Y.; software, Y.L. and L.Y.; validation, W.W., H.C. and C.H.; writing—original draft preparation, Y.L.; writing—review and editing, L.Y. and C.H.; funding acquisition, L.Y. All authors have read and agreed to the published version of the manuscript.

**Funding:** The research is supported by the National Natural Science Foundation of China under grant numbers U2141220, 116723 and 12102440.

**Conflicts of Interest:** The authors declare no conflict of interest.

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
