# Peer review of "Lagrange Optimization of Shock Waves for Two-Dimensional Hypersonic Inlet with Geometric Constraints"

_aerospace, doi:10.3390/aerospace9100625_

Round 1

Reviewer 1 Report

Dear authors, 

enclosed you can find the comments to your work.

Regard.

Reviewer 2 Report

An interesting optimization problem is described in the paper. I suggest some minor modifications which can improve the quality of the work:

* The chosen goal function is the logarithm of the total pressure recovery. The Authors should clarify what are the benefits in using the logarithm instead of the pressure recovery itself (better scaling?)

* The bibliography should be improved by adding more recent works on supersonic inlets

* The Authors should discuss in the Conclusion some limits of the proposed approach: even considering a planar 2D inlet, there are end walls which introduce 3D effects in some regions of the domain. Furthermore, shock waves can induce boundary layer separation which can strongly influence pressure recovery. I suggest to add comments on these problems in the Conclusion.

* The text should be checked in order to re-write some sentences which can bring confusion. For example, in the abstract the following sentence should be clarified: " The rule of equal intensity between internal and external compression shock waves would be broken by limiting the inlet length "

Round 2

Reviewer 1 Report

Dear authors.

In my opinion the work can be accepted for publication.

Regards.